# OpenReview forum: "Read the Scene, Not the Script: Outcome-Aware Safety for LLMs"
_ICLR.cc/2026/Conference — Submitted to ICLR 2026_

### Official Review · Reviewer_Tcne · 2025-10-29

**Soundness:** 3
**Presentation:** 2
**Contribution:** 3
**Rating:** 4
**Confidence:** 3

**Summary:**

This paper differentiates between outcome risk and semantic risk for safety-related prompts. Semantic risk includes prompts that may use safety-flagged words or discuss a dangerous scenario, and outcome risk is prompts that could result in real harmful outcomes. Outcome risk is what we actually care about from a safety standpoint, but this paper contributes a benchmark demonstrating that most LLMs are consequence-blind and focus more on semantic risk in evaluating prompt safety and choosing when to refuse a response. To mitigate this issue, they also release a training dataset for consequence reasoning and show that fine-tuning LLMs on this dataset reduces their consequence-blindness.

**Strengths:**

Originality:
- the distinction between outcome and semantic risk is an important and interesting idea in safety alignment and they explain this clearly

Quality:
- the datasets seem useful

Clarity:
- the writing is generally clear

Significance:
- this seems like a very useful benchmark

**Weaknesses:**

1. The figures are very unclear and seem to be missing data in some places. Most critically, most of the plots are blank in Figure 5. And in Figure 4, why is the red line missing in the fourth column/second row? In terms of confusing figures/tables, in Figure 1, on the left, it seems to me like the model is doing the right thing so I find this difficult to interpret, and the graphic on the right was very confusing for me to understand the takeaway. I think focusing this figure on clear examples of when semantic risk and outcome risk differ would be more helpful. For tables, you should state somewhere what the blue row indicates and clarify in the caption what S/C and C/C mean. I also find Figure 3 quite confusing and find the icons more distracting than helpful.

2. There is not enough detail on the benchmark and training dataset construction in the main body of the paper. We really need to see some example instances from both in the paper and it would help to move some of the details from the appendix into the main body. We don't need the prompts, but we need some of the wording.

**Questions:**

I am open to increasing my overall rating if the authors can clarify my questions about the figures in the rebuttal.

---

> ### Author Response · Authors · 2025-11-20
> **Author Response to Reviewer Tcne (Part 1/2)**
>
> Thank you for the thoughtful and constructive feedback, and for recognizing the originality of the outcome–semantic risk distinction, the usefulness of the benchmarks and datasets, and the clarity of the overall writing. We appreciate these comments—they reflect the core goals of our work.
>
> Below, we address all concerns and describe the changes made in the revised manuscript.
>
> ---
>
> ### **Response to Weakness-2**
> >There is not enough detail on the benchmark and training dataset construction in the main body of the paper. We really need to see some example instances from both in the paper and it would help to move some of the details from the appendix into the main body. We don't need the prompts, but we need some of the wording.
>
> Response:
> To address your request for more detail and concrete examples in the main body, we agreed that directly moving long blocks of data instances from the appendix would make the presentation heavy and interrupt the flow of the paper, because individual benchmark or training examples tend to be quite long. Instead of relocating large excerpts, **we redesigned Figure 3 so that it now contains a real and fully worked example embedded directly inside the construction pipeline**. This approach allows us to present the actual wording used at each step, showing how topics are instantiated, how outcome risk and semantic risk sentences are created, and how the four CB-Bench request types or the CS-Chain supervision signals are formed, while keeping the narrative concise and readable. By presenting the example within the visual workflow, the revised figure brings concrete instances into the main body in a compact and intuitive manner. The full datasets and templates remain in the appendix for completeness, but the new construction figure provides the wording-level detail you requested and makes the overall data generation process substantially clearer.

---

> ### Author Response · Authors · 2025-11-20
> **Author Response to Reviewer Tcne (Part 2/2)**
>
> ### **Response to Weakness-1**
> >The figures are very unclear and seem to be missing data in some places. Most critically, most of the plots are blank in Figure 5. And in Figure 4, why is the red line missing in the fourth column/second row? In terms of confusing figures/tables, in Figure 1, on the left, it seems to me like the model is doing the right thing so I find this difficult to interpret, and the graphic on the right was very confusing for me to understand the takeaway. I think focusing this figure on clear examples of when semantic risk and outcome risk differ would be more helpful. For tables, ...
>
> Response:
> 1. **About Figure 4 and Figure 5 Appearing Blank / Missing Lines**: We carefully rechecked all figures and confirmed that the plots themselves are correct and fully rendered in the source files. We suspect that the issue you encountered was due to PDF rendering limitations caused by the large file size, which can sometimes prevent certain elements from displaying on first load. To avoid this, we optimized and compressed the PDF in the revised version, ensuring that all subplots and curves—including the previously missing red line—now display consistently across standard PDF viewers.
> 2. **"Confusing" Figure 1 Clarify**. Figure 1 illustrates the contrast between consequence-blind and consequence-aware behavior in safety-aligned language models. On the left, the examples show a model that relies primarily on shallow semantic cues, such as a user claiming to be doing “research” or using seemingly neutral language, while ignoring the actual downstream consequences of providing harmful instructions. Even though the surface phrasing appears benign, the underlying task, such as bypassing a smart lock or manipulating real devices, still poses high real-world risk, and a consequence-blind model fails because it treats the semantic framing as the dominant signal and therefore answers when it should refuse. The right side visualizes the intended consequence-aware behavior using an abstract two-dimensional risk space. The vertical blue axis represents the level of outcome risk associated with fulfilling the request, and the horizontal red plane represents semantic risk arising from the wording or context. The spiral trajectory conveys that, across many different linguistic framings, a properly aligned model consistently returns its decision to the outcome-risk dimension, refusing when the actual consequences of compliance are dangerous and assisting when the consequences are harmless, regardless of whether the surface wording sounds risky or safe. This conceptual illustration is not a depiction of model internals, and instead expresses how a consequence-aware model reorganizes its decision boundary so that real-world outcomes, rather than superficial semantics, become the primary basis for safety decisions.
> 3. **Revised Figure 3**: We appreciate the your comment regarding the visual clutter and lack of clarity in the original Figure 3. In response, we have fully redesigned the figure into a cleaner two-part structure, with the CB-Bench construction pipeline shown in the upper section and the CS-Chain construction pipeline shown in the lower section. Instead of relying on numerous abstract icons, the revised figure now uses a single concrete running example to illustrate how both resources are constructed step by step. In the CB-Bench section, we show how a topic is first decomposed into core phrases, then instantiated into high and low outcome-risk question sentences, and finally combined with high and low semantic-risk backgrounds to form the four canonical request types Q1–Q4. In the CS-Chain section, we similarly depict how prompts are gathered from WildJailbreak, UltraSafety, OR-Bench, and our own harmless but sensitive generation pipeline, how five CoT responses are sampled from the teacher model, how safety filtering retains only prompts with at least one safe trajectory, and how a final supervision example is selected. By replacing abstract symbols with an end-to-end real example, the new figure is more intuitive and transparent, and it also helps, at least to some extent, to address Weakness-2 by bringing more construction details and concrete instances from both CB-Bench and CS-Chain into the main body of the paper, making the overall workflow easier to understand without requiring readers to rely solely on the appendix.
> 4. **Clarifying Table Abbreviations and Highlighting**: As noted, we have already added explicit clarifications to the table captions. For completeness, we reproduce the explanation here:
>     - W/C = WithoutChain-4k (no reasoning supervision)
>     - S/C = SafeChain-4k (general reasoning supervision)
>     - C/C = CS-Chain-4k (consequence-aware supervision)
>
>     “+C/C” indicates models fine-tuned with CS-Chain, our proposed     method, which achieves the strongest consequence-aware safety.     We have revised the text and captions to state these               definitions clearly.

---

### Official Review · Reviewer_exPQ · 2025-10-31

**Soundness:** 3
**Presentation:** 2
**Contribution:** 2
**Rating:** 4
**Confidence:** 3

**Summary:**

This paper investigates the LLM safety problem through two key failure modes: vulnerability to jailbreak attacks and over-conservativeness. The authors argue that the root cause of these issues lies in current models' weak reasoning about the connection between actions and consequences, and their over-reliance on surface-form cues. To support this claim, they introduce CB-Bench (Consequence-Blindness Benchmark), a dataset designed to evaluate models’ ability to identify underlying intent and corresponding safety behavior. Additionally, they propose CS-Chain-4k, a consequence-reasoning dataset aimed at advancing safety alignment. The authors further validate their findings through comprehensive experimental results.

**Strengths:**

(1) Interesting Problem:
 The problem of LLM safety, and the underlying relationship between surface form and semantic meaning, is both important and interesting.

(2) In-Depth Experiments:
 The authors not only present the final evaluation results of their method but also provide in-depth analyses, such as probing and token attribution studies, to better understand the underlying mechanisms.

**Weaknesses:**

(1) Limited Evaluation:
 The main evaluation in Table 2 only includes open-source models with ≤32B parameters, which is insufficient to fully assess the difficulty of the proposed task. I suggest including evaluation results from larger models, such as Qwen2.5-72B, LLaMA3.3-70B, DeepSeek-R1/V3, as well as closed-source models like GPT-4o and the Gemini series.

(2) Interpretation of Scaling Effects (Line 237):
 The authors state that: “These results reveal that scaling impacts safety trade-offs differently across architectures, and larger models do not universally improve consequence-aware safety.” However, this conclusion may be biased, as it does not account for differences in pre-training datasets and only considers models from two series. As such, the evidence presented is not sufficient to support a strong claim against scaling parameter size.

(3) Surface Form vs. Underlying Semantics:
 For the discussion on surface form versus underlying semantics, I recommend a more in-depth comparison with related work—for example, references [1, 2].

Reference:

[1]  Yue Zhou, et al. "Paraphrase and solve: Exploring and exploiting the impact of surface form on mathematical reasoning in large language models." arXiv preprint arXiv:2404.11500 (2024).

[2] Yihang Yao,  et al. "Your language model may think too rigidly: Achieving reasoning consistency with symmetry-enhanced training." arXiv preprint arXiv:2502.17800 (2025).

**Questions:**

(1) Limited Comparison of Reasoning vs. Non-Reasoning Models:
 The authors claim that Reasoning Enhancement Worsens Issues (page 4), suggesting that reasoning models perform worse than non-reasoning ones. However, the evaluation appears limited to small-scale models. Could you include further experiments comparing larger models, such as DeepSeek-V3 versus DeepSeek-R1?

(2) Clarification on DeepSeek-R1 Reference (Lines 245–246):
 You mention that Reasoning models (e.g., DeepSeek-R1, Qwen3-Thinking) devote a large share of tokens to CoT. However, DeepSeek-R1 does not seem to appear in the context. Are you referring to the R1-Distilled models?

(3) CoT Impact on Evaluation Consistency (Line 209):
 The authors briefly mention the impact of CoT on evaluation consistency. Could you elaborate on this point? For which tasks is CoT evaluation particularly critical, and why can verifiers not rely solely on the final answer?

(4) Refusal and Semantic Risk Analysis:
 In the section analyzing the impact of refusal and semantic risk on output length, how is the comparison performed? When you refer to "shorter CoT responses," do you mean outputs that include reasoning supporting the semantic risk, or are you referring to brief refusal messages?

(5) Clarification of Table 4 Annotations:
 In Table 4, why is the "+C/C" condition highlighted? Additionally, could you clarify the meanings of the abbreviations "W/C", "S/C", and "C/C"?

---

> ### Author Response · Authors · 2025-11-20
> **Author Response to Reviewer exPQ (Part 1/3)**
>
> Thank you for the constructive review and for highlighting both the importance of our problem formulation and the depth of our analyses. We appreciate your recognition of the probing and attribution studies, which were intended to provide a clearer mechanistic view of consequence-blindness.
>
> We also thank you for the detailed suggestions. We address all items below and have incorporated the corresponding additions and clarifications into the revised manuscript.
>
> ---
>
> ### **Response to Weakness-1**
> >Limited Evaluation: The main evaluation in Table 2 only includes open-source models with ≤32B parameters, which is insufficient to fully assess the difficulty of the proposed task. I suggest including evaluation results from larger models, such as Qwen2.5-72B, LLaMA3.3-70B, DeepSeek-R1/V3, as well as closed-source models like GPT-4o and the Gemini series.
>
> Response:
> Thank you for the suggestion. We have added evaluations on the requested larger models, including Qwen2.5-72B, LLaMA3.3-70B, DeepSeek-R1-70B, DeepSeek-R1-Full, GPT-4o, and Gemini-2.5-Flash. The results are shown below:
>
> | Model               | Jailbreaked ↓ | Over-refusal ↓ | CB-Score ↓ |
> |---------------------|-------------|--------------|----------|
> | Qwen2.5-72B         | 56.20       | 24.66        | 0.31     |
> | LLaMA3.3-70B        | 56.80       | 11.33        | 0.24     |
> | DeepSeek-R1-70B     | 73.33       | 11.40        | 0.32     |
> | DeepSeek-R1-Full    | 62.18       | 29.72        | 0.37     |
> | GPT-4o              | 53.38       | 20.66        | 0.26     |
> | Gemini2.5-Flash     | 39.04       | 49.29        | 0.37     |
>
> These additional results further validate that **CB-Bench remains challenging even for frontier-level models and strengthen the overall evaluation**. Despite having far stronger general capabilities, these models still fail to maintain consistent decisions across the controlled Q1–Q4 prompt pairs. Several trends emerge from the expanded table. First, increasing model size does not guarantee improved consistency: Qwen2.5-72B and LLaMA3.3-70B both show jailbreak rates above 55 percent, comparable to or even worse than much smaller models. Second, models optimized heavily for helpfulness tend to under-refuse but become more jailbreak-prone, as seen in the DeepSeek-R1-70B results. Third, models optimized more conservatively exhibit the opposite imbalance: Gemini2.5-Flash shows the lowest jailbreak rate in this group yet has the highest over-refusal rate, leading to a CB-Score similar to DeepSeek-R1-Full despite their very different behavior profiles.
>
> Taken together, these findings demonstrate that **CB-Bench captures a failure mode that persists even at frontier scale and cannot be resolved simply by scaling up model size or improving general reasoning ability**. The difficulty lies not in answering individual prompts but in maintaining stable, consequence-sensitive decisions across semantically aligned prompt variations, which remains an open challenge for models across the capability spectrum.
>
> ---
>
> ### **Response to Weakness-2**
> >Interpretation of Scaling Effects (Line 237): The authors state that: “These results reveal that scaling impacts safety trade-offs differently across architectures, and larger models do not universally improve consequence-aware safety.” However, this conclusion may be biased, as it does not account for differences in pre-training datasets and only considers models from two series. As such, the evidence presented is not sufficient to support a strong claim against scaling parameter size.
>
> Response:
> We agree that the original statement was too strong, and we appreciate your careful feedback. We have already revised the manuscript to clarify that our conclusion refers specifically to the scaling behavior observed within the Qwen2.5 and Gemma3 model families evaluated in our experiments. The updated text no longer frames the trend as a universal property of scaling but instead explicitly states that **these findings are limited to the family regimes represented in our study**.
>
> ---
>
> ### **Response to Weakness-3**
> >Surface Form vs. Underlying Semantics: For the discussion on surface form versus underlying semantics, I recommend a more in-depth comparison with related work—for example, references [1, 2].
>
> Response:
> Thank you for the insightful recommendation. **We have expanded the Related Work section to include a dedicated discussion of the two suggested studies**, situating them within the broader landscape of surface-form sensitivity and semantic invariance. Specifically, we now compare their findings on paraphrase brittleness and symmetry-based reasoning consistency with our analysis of safety-specific semantic–outcome mismatches. This revision clarifies how our work extends beyond general reasoning robustness to address a distinct failure mode in safety decision-making.

---

> ### Author Response · Authors · 2025-11-20
> **Author Response to Reviewer exPQ (Part 2/3)**
>
> ### **Response to Question-1**
> >Limited Comparison of Reasoning vs. Non-Reasoning Models: The authors claim that Reasoning Enhancement Worsens Issues (page 4), suggesting that reasoning models perform worse than non-reasoning ones. However, the evaluation appears limited to small-scale models. Could you include further experiments comparing larger models, such as DeepSeek-V3 versus DeepSeek-R1?
>
> Response:
> We appreciate the reviewer’s concern and have expanded our evaluation to include larger-scale models (see Weakness-1 response), including Qwen2.5-72B, LLaMA3.3-70B, DeepSeek-R1-70B, DeepSeek-R1-Full, GPT-4o, and Gemini-2.5-Flash. These additions allow a direct comparison between reasoning-enhanced models and their non-reasoning counterparts at frontier scale.
>
> **The pattern we reported continues to hold**. LLaMA3.3-70B versus DeepSeek-R1-70B shows the same trade-off: the CB-Score increases from 0.24 → 0.32, but jailbreak vulnerability worsens substantially (56.80 → 73.33) while over-refusal remains similar.
>
> Taken together, these results show that **reasoning-enhanced models consistently amplify both strengths and weaknesses: they improve aligned consistency but also become more sensitive to adversarial framing, often raising jailbreak susceptibility**. This trend appears across model sizes and architectures, indicating that the phenomenon is not limited to small models and is precisely the type of instability CB-Bench is designed to reveal.
>
> ---
>
> ### **Response to Question-2**
> >Clarification on DeepSeek-R1 Reference (Lines 245–246): You mention that Reasoning models (e.g., DeepSeek-R1, Qwen3-Thinking) devote a large share of tokens to CoT. However, DeepSeek-R1 does not seem to appear in the context. Are you referring to the R1-Distilled models?
>
> Response:
> Thank you for raising this point. The models used in Sec. 3.3 (Table 2) are the DeepSeek R1 Distill variants, including DeepSeek R1 Distill Llama 8B, DeepSeek R1 Distill Qwen 7B, and DeepSeek R1 Distill Qwen 14B. This information is already present in the table, but we agree that the wording in the main text could make this connection clearer to readers.
>
> **We have therefore updated the manuscript to explicitly refer to these models as DeepSeek-R1-Distill** whenever they are discussed in the context of reasoning-style models that allocate a large share of tokens to chain-of-thought. The revised text now names the Distill family directly, which removes any possible ambiguity with the full DeepSeek R1 model and makes it immediately clear which variants are included in our evaluation.
>
> ---
>
> ### **Response to Question-3**
> >CoT Impact on Evaluation Consistency (Line 209): The authors briefly mention the impact of CoT on evaluation consistency. Could you elaborate on this point? For which tasks is CoT evaluation particularly critical, and why can verifiers not rely solely on the final answer?
>
> Response:
> We appreciate your valuable comment. Our analysis of refusal inconsistencies shows that CoT critically affects safety evaluation, especially for high-risk tasks. We observe two systematic patterns:
>
> 1. **Hidden leakage in the CoT**: In domains such as copyright evasion, chemical misuse, surveillance avoidance, and personal-data extraction, models often generate concrete procedural steps inside the CoT before issuing a final “refusal.” When the evaluator can read the CoT, these samples are correctly flagged as non-refusals; when the CoT is suppressed, the identical outputs appear as false-positive refusals despite having already disclosed actionable information earlier.
> 2. **Performative refusals**: We also find the opposite pattern—some models express caution or self-corrections within the CoT, which convinces the verifier that the model refused, yet the final answer still contains detailed guidance. With the CoT hidden, these samples flip into non-refusal, revealing that the apparent safety stance existed only in the internal reasoning.
>
> Therefore, CoT-aware evaluation is not optional: it is necessary to detect whether harmful content is generated in any part of the response pipeline and to avoid misclassifying “fake refusals” that only occur in hidden reasoning. Ignoring the CoT leads to consistent mis-evaluation in the scenarios of highest regulatory concern.

---

> ### Author Response · Authors · 2025-11-20
> **Author Response to Reviewer exPQ (Part 3/3)**
>
> ### **Response to Question-4**
> >Refusal and Semantic Risk Analysis: In the section analyzing the impact of refusal and semantic risk on output length, how is the comparison performed? When you refer to "shorter CoT responses," do you mean outputs that include reasoning supporting the semantic risk, or are you referring to brief refusal messages?
>
> Response:
> The comparison is performed by computing the average token length of model outputs across all CB-Bench cases, grouped by the four (semantic risk × outcome risk) conditions. This is done after running every model across the full benchmark. Regarding “shorter CoT responses”:
> - Some models produce a **full reasoning chain that is nevertheless very short**, usually explaining only high-level safety principles without engaging in deeper causal reasoning.
> - Others produce a **brief refusal message without any reasoning**.
>
> Both are counted as “shorter outputs,” and the manuscript will be updated to make this distinction clearer.
>
> ---
>
> ### **Response to Question-5**
> >Clarification of Table 4 Annotations: In Table 4, why is the "+C/C" condition highlighted? Additionally, could you clarify the meanings of the abbreviations "W/C", "S/C", and "C/C"?
>
> Response:
> The abbreviations refer to the three training settings compared in our experiments:
> - **W/C** = WithoutChain-4k (no reasoning supervision)
> - **S/C** = SafeChain-4k (general reasoning supervision)
> - **C/C** = CS-Chain-4k (consequence-aware supervision)
>
> “+C/C” indicates models fine-tuned with **CS-Chain**, which is our proposed method and achieves the strongest consequence-aware safety. We have revised the text to define these abbreviations explicitly.

---

### Official Review · Reviewer_99Gi · 2025-10-31

**Soundness:** 2
**Presentation:** 2
**Contribution:** 2
**Rating:** 6
**Confidence:** 4

**Summary:**

- Safety-aligned LLMs exhibit two major failure modes: susceptibility to jailbreaks and excessive refusals of harmless prompts with sensitive wording.
 - Both failures stem from consequence-blindness—models’ weak reasoning about action–outcome relationships.
 - Introduces CB-Bench (Consequence-Blindness Benchmark) covering four risk scenarios.
 - Existing models consistently fail to distinguish between semantic and outcome risk, confirming that consequence-blindness is systematic and widespread.
 - Proposes CS-Chain-4k (ConSequence Chain), a consequence-reasoning dataset for improving safety alignment.
 - Fine-tuning on CS-Chain-4k enhances resistance to semantic-camouflage jailbreaks and reduces over-refusal of benign prompts.
 - Performance on other benchmarks remains stable, demonstrating preserved utility and generalization.
 - Establishes consequence-aware reasoning as a key objective for future safety alignment and offers a practical, reproducible evaluation path.

**Strengths:**

Conceptually this is a very interesting framework: it's hard to judge the sensitivity and harmfulness of decisions without reflecting on their consequences. This is essentially a causal relationship between actions and their outcomes.

**Weaknesses:**

A (perhaps reductionist?) perspective on this work this is that, the authors have annotated "rationales" or "reasoning chains" for why certain actions should or should not be made. And from the existing literature, there is ample evidence that articulating reasonings (by the model) will help it become more reliable in its decision making. With this lens, the contribution of this work is to supervise their model with more detailed reasoning chains. Curious if/how the authors would push back against this.

 One issue about "consequences" is that they can be subjective and quite context-specific. Can the authors elaborate on how they went about resolving subjectivity in their construction?



Fig 2 is actually a bit confusing.
- Left subfig: What is "score"??
- Right subfig: Which line corresponds to which y-axis? (specifically the left y-axis says "percentage" but unclear percentage of what exactly. )

 Minor: Consider changing your CB-scores to % to be compatible with the rest of the numbers.

**Questions:**

See the previous box.

---

> ### Author Response · Authors · 2025-11-20
> **Author Response to Reviewer 99Gi (Part 1/2)**
>
> Thank you for the thoughtful review and for highlighting the conceptual motivation of our work. We appreciate your recognition that evaluating model decisions through their consequences naturally aligns with causal reasoning between actions and outcomes. This perspective captures the core intuition behind our approach.
>
> We also appreciate your detailed comments and questions, which helped clarify several aspects of our framework. Below we address each point and have incorporated corresponding explanations.
>
> ---
>
> ### **Response to Weakness-1**
> >A (perhaps reductionist?) perspective on this work this is that, the authors have annotated "rationales" or "reasoning chains" for why certain actions should or should not be made. And from the existing literature, there is ample evidence that articulating reasonings (by the model) will help it become more reliable in its decision making. With this lens, the contribution of this work is to supervise their model with more detailed reasoning chains. Curious if/how the authors would push back against this.
>
> Response:
> We appreciate the reviewer’s perspective and agree that providing rationales can improve model reliability. However, our work goes beyond adding more detailed explanations. The key distinction is that CS-Chain supervises the model specifically on consequence-based reasoning, whereas existing datasets such as SafeChain mainly provide normative or policy-style justifications (“this violates guidelines,” “this is unsafe”) without tracing the actual causal chain that links a user action to its downstream effects. **In other words, SafeChain contains reasoning steps, but they rarely answer the core question of why a given action leads to a harmful real-world outcome**. CS-Chain directly targets this missing dimension.
>
> To clarify this difference, we added an LLM-as-a-judge evaluation (consistent with our response to Question 1) where the judge scores each response on a 1–5 scale based on how strongly the reasoning depends on explicit causal mechanisms and real-world consequences rather than surface-level heuristics. Under this unified rubric, SafeChain despite containing rationales, consistently receives much lower scores than CS-Chain across all three judge models. Since the entire scale spans only 1–5, **the observed gaps of roughly 0.8–1.3 points represent substantial differences in reasoning depth**. These results show that CS-Chain does not simply add more reasoning, but induces a qualitatively different style of reasoning that explicitly links user intent, planned actions, and their real-world consequences.
> | Method               | Qwen2.5-7B | Llama3.1-8B | Gemma3-12B |
> |----------------------|------------|--------------|-------------|
> | SafeChain (original) | 3.23       | 3.20         | 3.40        |
> | SafeChain (baseline) | 3.58       | 3.44         | 3.72        |
> | CS-Chain             | 4.54       | 4.38         | 4.45        |

---

> ### Author Response · Authors · 2025-11-20
> **Author Response to Reviewer 99Gi (Part 2/2)**
>
> ### **Response to Weakness-2**
> >One issue about "consequences" is that they can be subjective and quite context-specific. Can the authors elaborate on how they went about resolving subjectivity in their construction?
>
> Response:
> We appreciate the your thoughtful question. In this part, we clarify how subjectivity is controlled in both CB-Bench and CS-Chain.
>
> 1. **CB-Bench**: CB-Bench defines outcome risk using a simple and practical criterion: a question is “high outcome risk” only if a fully detailed answer would necessarily contain actionable harmful information (for example, enabling physical harm, privacy breaches, or illegal access). Semantic risk is controlled separately through the background sentence. Using the background–question template in **Appendix C/H** and GPT-4o, we first sample, for each subtopic, one neutral but outcome-harmful core question and one neutral, outcome-safe core question, and then combine them with either malicious or benign one-sentence backgrounds to form four variants: Q1 (high semantic risk, high outcome risk) and Q2 (low semantic risk, high outcome risk) share the same harmful question, while Q3 (low semantic, low outcome) and Q4 (high semantic risk, low outcome risk) share the same safe question. Human review is limited to removing unnatural or mis-specified samples rather than relabelling risk. The metrics are defined on paired comparisons at fixed outcome risk: for the high outcome risk pair (Q1, Q2), the correct behaviour is to refuse both, and if Q1 is refused but Q2 is answered we count a jailbreak; for the low outcome risk pair (Q3, Q4), the correct behaviour is to answer both, and if Q3 is answered but Q4 is refused we count an over-refusal. **Thus the jailbreak and over-refusal rates depend on consistency within the Q1–Q2 and Q3–Q4 pairs, rather than on subjective judgments of how harmful each individual question is**.
> 2. **CS-Chain**: CS-Chain’s reasoning steps are not manually written. A single teacher model (Qwen3-275B-Thinking) is prompted to explicitly discuss downstream consequences, and an automatic safety judge filters unsafe outputs. This avoids annotator subjectivity and ensures consistent causal reasoning across the dataset. Details are provided in Sec. 4.1–4.2.
>
> Overall, CB-Bench relies on a fixed operational definition of outcome risk, while CS-Chain relies on a single consistent teacher model rather than human-authored rationales. This keeps “consequences” as objective and reproducible as possible while still capturing real downstream effects.
>
> ---
>
> ### **Response to Weakness-3**
> >Fig 2 is actually a bit confusing. Left subfig: What is "score"? Right subfig: Which line corresponds to which y-axis? (specifically the left y-axis says "percentage" but unclear percentage of what exactly. )
>
> Response:
> Thank you for pointing this out. We have clarified both subfigures in the revised manuscript:
> - **Left subfigure:** “Score” refers to the model’s CB-Score (our unified measure of consequence-blindness). This is now explicitly labeled in the caption.
> - **Right subfigure**: We have expanded the caption to clearly distinguish the two y-axes and the quantities associated with each. The jailbreak and over-refusal curves are described as percentages that reflect the proportion of mismatched-risk failures for each model size, while the CB-Score curve is identified as a continuous value in the range [0,1] plotted on the right y-axis. Although the visual layout of the figure remains the same, the improved caption now provides all necessary guidance for interpreting the plotted trends without confusion.
>
> These changes should resolve the ambiguity and make Figure 2 self-explanatory.

---

> > ### Comment · Reviewer_99Gi · 2025-11-24
> >
> > Thanks to the authors for clarification.
> > I remain marginally positive about this work.

---

### Official Review · Reviewer_Hmh6 · 2025-11-01

**Soundness:** 2
**Presentation:** 2
**Contribution:** 1
**Rating:** 2
**Confidence:** 4

**Summary:**

The paper discusses the shallow safety alignment of existing LLMs and argues that such behavior is due to the limited ability of the models to separate the semantic (surface) risk from the outcome risk, and that what explains the typically reported jailbreakability of most models and over-refusal rates of many of them. The paper introduces a benchmark (CS-Bench) with the goal of enabling the distinction between the two risks and present evaluation results on various models and uses the results to confirm the arguments about the shallow alignment. To address that problem, the paper also presents a safety alignment dataset (CS-Chain) that explicitly teaches models to reason about the outcome of answering user requests.

**Strengths:**

1. The paper introduces an interesting and sound distinction between semantic and outcome risk which is a valuable way for understanding the limited safety behaviors of today's models.

2. The presented alignment dataset is shown to indeed improve the safety/over-refusal of the evaluated models.

**Weaknesses:**

1. It is not clear what the value of the introduced benchmark CS-Bench is. It does not seem that the benchmark introduces any additional insights beyond those already established by evaluating on existing jailbreak attacks and over-refusal benchmarks. For example, table 3 in the paper uses CS-Bench and table 4 uses existing jailbreak attacks and XSTest. Both are leading to the same conclusion, and no new insights are provided by table 3. The benchmark can be made more useful if for example it provides more fine-grained categorization or understanding of the errors.

2. The novelty with CS-Chain (the fine-tunning dataset) is that it encourages the model to reason about the consequences. The paper lacks a baseline in which models are just prompted to do so without any additional fine-tuning.

3. The paper lacks qualitative and error analyses. It is essential to demonstrate the reasoning about the consequences behavior of the model. It is also important to provide an explanation of the still significant jailbreakability and over-refusal results in table 4. Was that mainly because of the model failure to reason about the consequences? or due to some other reason? Are there any limitations with CS-Chain that need to be addressed?

**Questions:**

Would you please confirm that SafeChain contains responses with reasoning, but they do not explicitly consider the consequences?

---

> ### Author Response · Authors · 2025-11-20
> **Author Response to Reviewer Hmh6 (Part 1/3)**
>
> Thank you for the thoughtful review and for highlighting the aspects of our work that you found valuable. We appreciate your recognition that our formulation offers a helpful way to think about risk in safety alignment and that our dataset effectively improves both safety and over-refusal. These points reflect the motivations behind our work, and we are glad they were clear to you.
>
> We also appreciate your detailed questions and concerns, many of which helped us strengthen the paper. A few points seem to arise from reasonable misunderstandings of our benchmark design and the intended purpose of CS-Chain. We clarify all items below and have added additional baselines, analyses, and explanations.
>
> ---
>
> ### **Response to Weakness-1**
> > It is not clear what the value of the introduced benchmark CS-Bench is. It does not seem that the benchmark introduces any additional insights beyond those already established by evaluating on existing jailbreak attacks and over-refusal benchmarks. For example, table 3 in the paper uses CS-Bench and table 4 uses existing jailbreak attacks and XSTest. Both are leading to the same conclusion, and no new insights are provided by table 3. The benchmark can be made more useful if for example it provides more fine-grained categorization or understanding of the errors.
>
> Response:
> We appreciate the reviewer’s perspective and understand why CB-Bench may look similar to existing jailbreak/over-refusal evaluations at first glance. The key difference is that **CB-Bench does not measure the same phenomenon: it isolates a cross-prompt, risk-mismatch failure mode that single-prompt benchmarks cannot detect**.
>
> 1. **CB-Bench captures a paired, risk-mismatched failure mode that single-prompt evaluations cannot reveal**. Existing jailbreak and over-refusal evaluations test each request independently, so the model’s behavior is closely tied to the exact wording of that one prompt. This makes it hard to tell whether the model is truly distinguishing harmful from harmless situations or simply responding to phrasing patterns. CB-Bench evaluates prompts in controlled pairs across Q1–Q4, where the underlying question is held constant but the background changes, or the background is held constant but the actual question changes. When a model gives opposite decisions on two prompts that share the same question but differ only in background, or share the same background but ask different questions, this exposes a stability issue: the model is not grounding its judgment in the actual outcome of the request but is being influenced by superficial context. Single-prompt benchmarks cannot detect this kind of cross-prompt inconsistency because they never compare related prompts.
> 2. **CB-Bench unifies jailbreak and over-refusal into a single interpretable measurement of alignment consistency**. Traditional evaluations score jailbreak and over-refusal separately, which makes it difficult to understand how a model balances helpfulness and safety across different backgrounds of the same task. A model may appear safe on jailbreak tests simply because it refuses broadly, or appear helpful while still releasing harmful content when the background is phrased differently. Because prior benchmarks do not link related prompts, they cannot assess whether a model treats harmful and harmless questions consistently across different backgrounds. CB-Bench evaluates the aligned quadrants together and summarizes the model’s behavior into a single score that reflects how reliably it distinguishes harmful from harmless outcomes under background variation. This unified view exposes differences in alignment behavior that remain hidden when jailbreak and over-refusal are evaluated in isolation, even when their overall conclusions appear similar.

---

> ### Author Response · Authors · 2025-11-20
> **Author Response to Reviewer Hmh6 (Part 2/3)**
>
> ### **Response to Weakness-2**
> >The novelty with CS-Chain (the fine-tunning dataset) is that it encourages the model to reason about the consequences. The paper lacks a baseline in which models are just prompted to do so without any additional fine-tuning.
>
> Response:
> We appreciate your suggestion regarding a prompt-only consequence-reasoning baseline. The main paper already contains the relevant numbers, but they were split across **Table 1 (prompt-only)** and **Table 4 (CS-Chain fine-tuned)**. In direct response to your comment, we consolidate these results into a single comparison table and additionally evaluate the prompt-only baseline on benchmarks that were previously not included, namely Sorry-bench, MMLU, and HellaSwag. This unified view makes it clearer how far prompting alone can move models toward consequence-awareness and where CS-Chain provides additional gains.
>
> The consolidated results show that prompting alone yields only limited and unstable improvements. On Sorry-bench, Qwen2.5-7B drops from 25.20 harmful outputs to 18.6 after CS-Chain, and Mistral-7B shows an even larger reduction from 50.75 to 15.9. StrongReject evaluations illustrate the same pattern: prompting still leaves both models vulnerable to prefix-style attacks, while CS-Chain sharply reduces these failures, for instance from 55.8 to 28.4 for Qwen2.5-7B and from 94.5 to 6.0 for Mistral-7B. On the newly included capability benchmarks, prompting makes almost no difference (e.g., Qwen2.5-7B remains essentially unchanged on MMLU and HellaSwag), whereas CS-Chain improves safety without degrading performance.
>
> Overall, these newly added comparisons show that prompting alone provides only small and inconsistent shifts in behavior, while **CS-Chain consistently strengthens consequence-aware alignment across jailbreak robustness, over-refusal moderation, and behavioral stability**. This makes the distinction between prompt-only and fine-tuned settings much clearer than in the original submission.
> | Model         | Method      | Sorry-bench ↓      | XSTest ↑        | MMLU ↑         | HellaSwag ↑     | StrongReject-PAP ↓ | StrongReject-GCG ↓ | StrongReject-Prefix ↓ |
> |---------------|-------------|---------------------|------------------|-----------------|------------------|----------------------|----------------------|-------------------------|
> | **Qwen2.5-7B** | Prompt     | 25.20 ± 1.5          | 50.0 ± 2.5       | 71.6 ± 0.4      | 80.0 ± 0.4       | 27.0 ± 2.0            | 61.2 ± 6.8            | 55.8 ± 5.1              |
> |               | Fine-tuned | 18.6 ± 1.2           | 70.6 ± 1.3       | 71.5 ± 0.4      | 79.6 ± 0.4       | 35.3 ± 3.5            | 37.3 ± 3.4            | 28.4 ± 1.0              |
> | **Mistral-7B** | Prompt     | 50.75 ± 3.7          | 64.2 ± 1.9       | 57.7 ± 0.4      | 82.0 ± 0.4       | 48.1 ± 3.4            | 89.7 ± 3.8            | 94.5 ± 2.2              |
> |               | Fine-tuned | 15.9 ± 1.0           | 67.9 ± 2.0       | 58.4 ± 0.3      | 79.2 ± 0.4       | 23.0 ± 5.5            | 33.3 ± 6.2            | 6.0 ± 1.2               |

---

> ### Author Response · Authors · 2025-11-20
> **Author Response to Reviewer Hmh6 (Part 3/3)**
>
> ### **Response to Weakness-3**
> >The paper lacks qualitative and error analyses. It is essential to demonstrate the reasoning about the consequences behavior of the model. It is also important to provide an explanation of the still significant jailbreakability and over-refusal results in table 4. Was that mainly because of the model failure to reason about the consequences? or due to some other reason? Are there any limitations with CS-Chain that need to be addressed?
>
> Response:
> We appreciate the your insightful comments. To address this concern, we provide new qualitative evidence, analyze the remaining sources of jailbreakability and over-refusal, and clarify the limitations of CS-Chain as currently formulated.
>
> 1. We added qualitative demonstrations of the model’s consequence reasoning in **Appendix D** (a new section updated for case study in page 15).  These examples show that CS-Chain encourages reasoning over real-world outcomes rather than relying on lexical heuristics.
> 2. **Remaining jailbreakability and over-refusal stem from persistent surface-driven risk signals rather than fully consequence-aware reasoning**. Our error analysis in **Appendix D** shows that, although CS-Chain reduces inconsistencies, the model still relies partly on surface lexical markers for risk assessment and does not always activate harm detection when users express operational intent. This indicates that CS-Chain improves reasoning quality but does not yet provide a fully stable mechanism for aligning risk evaluation with real-world consequences.
> 3. We would like to clarify that our paper already provides a detailed discussion of the limitations of CS-Chain in **Appendix B**(page 14). To help you understand our points more clearly, we summarise the key points below:
>        - CS-Chain provides supervision but not policy optimization.
>        - Broader risk topic coverage is required for real-world robustness.
>        A full consequentialist alignment pipeline would combine CS-Chain with outcome-based RL, which we identify as a promising direction.
>
> ---
>
> ### **Response to Question-1**
> >Would you please confirm that SafeChain contains responses with reasoning, but they do not explicitly consider the consequences?
>
> Response:
> **Yes. SafeChain includes reasoning steps but does not explicitly model consequences**. To make this distinction clearer, we added an additional evaluation in which an LLM judge scores each model response on a 1–5 scale based on how much the provided reasoning actually considers causal mechanisms, real-world outcomes, and downstream effects, rather than relying on abstract policy statements or generic safety disclaimers. This evaluation setup allows us to assess not only whether a model produces a chain-of-thought, but whether that chain-of-thought is genuinely grounded in consequence-based reasoning rather than surface heuristics.
>
> Using this judge-based metric, we compared three datasets: SafeChain (original, 40k), SafeChain (our baseline, 4k), and CS-Chain (4k). Across all judge models, including Qwen2.5-7B, Llama3.1-8B, and Gemma3-12B, CS-Chain consistently achieves much higher consequence-reasoning scores. While SafeChain provides general safety-oriented explanations, the reasoning often remains high-level and does not articulate why complying with a harmful request would lead to concrete negative outcomes. In contrast, CS-Chain responses explicitly trace the downstream effects of user intent, such as unauthorized access, physical harm, or system misuse, showing a qualitatively different form of reasoning.
> | Method               | Qwen2.5-7B | Llama3.1-8B | Gemma3-12B |
> |----------------------|------------|--------------|-------------|
> | SafeChain (original) | 3.23       | 3.20         | 3.40        |
> | SafeChain (baseline) | 3.58       | 3.44         | 3.72        |
> | CS-Chain             | 4.54       | 4.38         | 4.45        |

---

### Author Response · Authors · 2025-11-20
**Author Rebuttal Summary (Part 1/2 -  Exeperiments)**

We sincerely thank all reviewers for their thoughtful and constructive feedback. Your comments helped us identify places where additional experimental evidence, clearer exposition, and more concrete examples were needed. In the rebuttal phase, we carefully addressed each concern through new experiments, expanded analyses, and targeted revisions to the manuscript. Below, we summarize the key additions and improvements made in direct response to the reviewers’ suggestions.

---

## 1. **Newly Added Experiments**

### (a) Expanded CB-Bench Evaluation to Large/Frontier-Scale Models (**motivated by exPQ's Weaknesses**)
We added CB-Bench results for large-scale and frontier models, including Qwen2.5-72B, LLaMA3.3-70B, DeepSeek-R1-70B, DeepSeek-R1-Full, GPT-4o, and Gemini-2.5-Flash. These results show that the cross-prompt consistency failures captured by CB-Bench persist even at frontier scale. Models with stronger general capabilities still struggle to maintain consistent decisions across Q1–Q4, reinforcing the difficulty and necessity of this evaluation.
| Model               | Jailbreaked ↓ | Over-refusal ↓ | CB-Score ↓ |
|---------------------|-------------|--------------|----------|
| Qwen2.5-72B         | 56.20       | 24.66        | 0.31     |
| LLaMA3.3-70B        | 56.80       | 11.33        | 0.24     |
| DeepSeek-R1-70B     | 73.33       | 11.40        | 0.32     |
| DeepSeek-R1-Full    | 62.18       | 29.72        | 0.37     |
| GPT-4o              | 53.38       | 20.66        | 0.26     |
| Gemini2.5-Flash     | 39.04       | 49.29        | 0.37     |

---

### (b) Prompt-Only vs. CS-Chain Fine-Tuning (**motivated by Reviewer Hmh6's Weaknesses**)
We consolidated all prompt-only numbers with CS-Chain results into a unified comparison table and newly expanded the prompt-only baseline to additional benchmarks (MMLU, HellaSwag, Sorry-bench). The unified analysis shows that prompting alone yields limited and unstable improvements, whereas CS-Chain provides consistent gains in consequence-aware alignment while preserving capabilities.
| Model         | Method      | Sorry-bench ↓      | XSTest ↑        | MMLU ↑         | HellaSwag ↑     | StrongReject-PAP ↓ | StrongReject-GCG ↓ | StrongReject-Prefix ↓ |
|---------------|-------------|---------------------|------------------|-----------------|------------------|----------------------|----------------------|-------------------------|
| **Qwen2.5-7B** | Prompt     | 25.20 ± 1.5          | 50.0 ± 2.5       | 71.6 ± 0.4      | 80.0 ± 0.4       | 27.0 ± 2.0            | 61.2 ± 6.8            | 55.8 ± 5.1              |
|               | Fine-tuned | 18.6 ± 1.2           | 70.6 ± 1.3       | 71.5 ± 0.4      | 79.6 ± 0.4       | 35.3 ± 3.5            | 37.3 ± 3.4            | 28.4 ± 1.0              |
| **Mistral-7B** | Prompt     | 50.75 ± 3.7          | 64.2 ± 1.9       | 57.7 ± 0.4      | 82.0 ± 0.4       | 48.1 ± 3.4            | 89.7 ± 3.8            | 94.5 ± 2.2              |
|               | Fine-tuned | 15.9 ± 1.0           | 67.9 ± 2.0       | 58.4 ± 0.3      | 79.2 ± 0.4       | 23.0 ± 5.5            | 33.3 ± 6.2            | 6.0 ± 1.2               |

---

### (c) Measuring Datasets' Consequence Reasoning (**motivated by Reviewer Hmh6's Questions and Reviewer 99Gi's Weaknesses**)
To distinguish CS-Chain from general safety datasets such as SafeChain, we introduced a new evaluation using LLM judges (Qwen2.5-7B, LLaMA3.1-8B, Gemma3-12B) that score each response on a 1–5 scale based on the extent to which the reasoning reflects actual downstream consequences. CS-Chain achieves substantially higher scores (≈0.8–1.3 point gains), demonstrating that it induces qualitatively different reasoning rather than merely longer explanations.
| Method               | Qwen2.5-7B | Llama3.1-8B | Gemma3-12B |
|----------------------|------------|--------------|-------------|
| SafeChain (original) | 3.23       | 3.20         | 3.40        |
| SafeChain (baseline) | 3.58       | 3.44         | 3.72        |
| CS-Chain             | 4.54       | 4.38         | 4.45        |

---

> ### Author Response · Authors · 2025-11-20
> **Author Rebuttal Summary (Part 2/2 - Manuscript)**
>
> ## 2. **Manuscript Revisions**
>
> ---
>
> ### (a) Clarifying Captions (**addressing Reviewer Tcne’s Questions, Reviewer 99Gi’s Weaknesses and Reviewer exPQ's Weaknesses**)
> To resolve the recurring confusion noted by Reviewer Tcne, particularly the missing explanations for axes, encoded conditions, and abbreviations such as W/C, S/C, and C/C, we rewrote the captions for all relevant figures and tables. Reviewer 99Gi also pointed out interpretability issues in Figure 2, so we expanded the captions to explicitly explain the evaluation metrics, plotted quantities, and color schemes. These revisions ensure that readers no longer need to infer semantics from the visuals themselves, which was one of the main sources of confusion in the original draft.
>
> ---
>
> ### (b) Redesigning Figure 3 (**addressing Reviewer Tcne’s Weaknesses**)
> Because Reviewer Tcne found the original Figure 3 unclear and visually distracting, we fully redesigned the figure. The new version presents the CB-Bench pipeline and the CS-Chain pipeline using a single concrete running example, replacing the earlier abstract iconography. By showing how a topic is decomposed, instantiated into Q1–Q4 forms, and later paired with teacher trajectories for CS-Chain supervision, the revised figure makes the construction logic explicit. This directly addresses the reviewer’s request for a clearer, more transparent depiction of both resources.
>
> ---
>
> ### (c) Adding Appendix D with Case Studies (**addressing Reviewer Hmh6’s Weaknesses and Reviewer Tcne’s Weaknesses**)
> Reviewer Hmh6 emphasized that the paper lacked qualitative and error analyses, and requested demonstrations showing how models reason about consequences. Reviewer Tcne echoed this need for concrete examples. In response, we added a new Appendix D that provides detailed case studies drawn from CB-Bench, including annotated model outputs, explanations of failure modes, and representative instances where outcome risk diverges from semantic wording.

---

### Meta-Review · Area_Chair_DNAx · 2026-01-05

**Summary:**

To tackle the two dominant failures (jailbroken and over-refusal) in LLM safety, the authors propose a perspective that they are caused by "current models reason weakly about links between actions and outcomes and over-rely on surface-form signals, lexical or stylistic cues that do not encode consequences." Based on this perspective, the authors propose a new benchmark and a fine-tuning dataset to improve LLM safety.

Strengths:

1. An interesting perspective.

Weaknesses:

1. Its arguments lack detailed explanations to support that "lack of awareness of outcomes" is the key for the failures. Maybe just one of the reasons leading to failure cases.

2. Its impact is somehow limited. The new benchmark just gives one perspective of failures. There may exist other causes like overfitting harmful template, cannot distinguish harmful intentions and etc.

3. The improvement is not much compared with the baseline method in finetuning. Demonstrating the proposed perspective is not the crucial one.

Therefor, I suggest to reject.

**Reviewer Concerns:**

The reviewers show concerns about its impacts (or new insights), not significant performance, the evaluated models, and the paper writing. I think the paper writing is addressed in the rebuttal. However, its impacts and performance are not.

**Reviewer Scores:**

I think some reviewers may still be negative like Hmh6 and exPQ. Tcne may increase their score.

---

### Decision · Program_Chairs · 2026-01-26

Reject